# Grid-forming control strategies for blackstart by offshore wind power plants

Anubhav Jain[1], Jayachandra N. Sakamuri[2], and Nicolaos A. Cutululis[1]

[1]DTU Wind Energy, Technical University of Denmark, 4000 Roskilde, Denmark
[2]Vattenfall A/S, 6000 Kolding, Denmark

**Correspondence:** Anubhav Jain (anub@dtu.dk)

**Abstract.** Large-scale integration of renewable energy sources with power-electronic converters is pushing the power system closer to its dynamic stability limit. This has increased the risk of wide-area blackouts. Thus, the changing generation profile in the power system necessitates the use of alternate sources of energy such as wind power plants, to provide blackstart services in the future. However, this requires *grid-forming* and not the traditionally prevalent *grid-following* wind turbines. This paper introduces the general working principle of grid-forming control and examines four of such control schemes. To compare their performance, a simulation study has been carried out for the different stages of energization of onshore load by an HVDC-connected wind power plant. Their transient behaviour during transformer inrush, converter pre-charging and de-blocking, and onshore block-load pickup, has been compared and analysed qualitatively to highlight the advantages and disadvantages of each control strategy.

## Nomenclature

| | |
|---|---|
| dPLL | Distributed PLL-based Control |
| DPC | Direct Power Control |
| DRU | Diode Rectifier Unit |
| GFL | Grid Following Control |
| GFM | Grid Forming Control |
| GSC | Grid Side Converter |
| (O)WPP | (Offshore) Wind Power Plant |
| PCC | Point of Common Coupling |
| PEC | Power Electronic Converter |
| PIR | Pre Insertion Resistor |
| PIT | Pre Insertion Time |
| PLL | Phase Locked Loop |
| PSC | Power Synchronization Control |

| VSC | Voltage Source Converter |
| VSG | Virtual Synchronous Generator Control |
| WT | Wind Turbine |

## 1   Introduction

Environmental problems like global warming, coupled with increasing fuel prices and the global drive towards sustainable development with energy security, has accelerated the integration of renewable energy sources into power systems all around the world. Many countries have set out several energy strategies for a more secure, sustainable and low-carbon economy like the European Union's (EU) 2018 (RED II) directive on the promotion of the use of energy from renewable sources that sets an overall goal across the EU for a 32% share of renewables in the total energy consumption by 2030. Among the different renewable energy sources, wind energy has seen a rapid growth in the installed capacity worldwide, from about 6.1 GW in 1996 to about 591.5 GW in 2018 (Tavner, 2012), showing huge promise as one of the major electricity sources in the future.

High volume integration of renewable energy into the power system makes it harder to maintain reliability and stability of power supply in the grid due to introduction of variable power flows and thus complicating grid operation (De Boeck et al., 2016). Moreover, the decrease in reactive power reserve due to replacement of conventional synchronous generation destabilizes the long-distance transmission corridors between load-centres and large-scale renewable energy systems—such as offshore wind power plants (OWPP)—during system contingencies (Sarkar et al., 2018). Additionally, inertial decoupling from the grid by the power electronic converter (PEC) interface results in decreased transient stability, increasing the risk of wide-area blackouts, especially in strongly linked networks (De Boeck et al., 2016). For example, as per Australian Energy Market Operator, the failure of WPP owners to comply with performance requirements to ride through major disruptions and disturbances led to blackout of the South Australia system in 2017, affecting about 850,000 people and causing large scale disruption to their livelihood and the economy. Another very recent case is the unexpected reduction of 737 MW from Hornsea 1 OWPP in the UK, that is cited to be one of the main causes of the system failure in August 2019, affecting about 1 million customers and causing travel chaos in and around London, according to the technical report by National Grid (2019a).

### 1.1   The changing paradigm

Traditionally blackstart service has been provided mainly by coal or gas-fired generators and pumped-hydro storage due to their capability to meet all the technical requirements (Elia, 2018; National Grid, 2019b). However, due to the societal decarbonisation aims, rising fuel costs coupled with ageing assets and decreasing load factors, large conventional generation plants are being phased out in favour of renewables and non-traditional technologies which increases the cost of warming-up large generators, and consequently of blackstart services (National Grid, 2019b). Since maintaining the status quo for blackstart and restoration is not an option, considerable changes are required to facilitate the participation of alternate sources like renew-

able energy and non-traditional technologies in the blackstart-market given the modern evolving energy landscape. Elia and National Grid for example, have recently confirmed that there is a potential to open up the delivery of blackstart-service to interconnectors, sites with trip to houseload operation and aggregated units including variable generation (like wind, solar), especially with support from energy storage systems.

Blackstart and islanding operation requirements have been included as options for WPPs in the ENTSO-E network codes, where the relevant system operator is allowed to request these functions to support grid-recovery (Göksu et al., 2017). Driven by grid codes, state-of-the-art wind turbines (WT) are already capable of providing some services that are a part of the restoration process—e.g. Fast Frequency Response and Low Voltage Ride Through (LVRT)—and are expected to deliver more advanced requirements like Inertia Emulation, Power Oscillation Damping and Reactive Current Injection, which are increasingly being demanded by grid-codes (Jain et al., 2019). This is possible due to the advanced functionalities of the full-scale PEC interface of modern WTs, as mentioned in Chen et al. (2009). Seca et al. (2013) show that WTs owing to their fast start-up times, can be included earlier in the restoration process to provide reactive-power support and pickup load, thus decreasing the impact of a blackout event by reducing the restoration time and unserved load. However, connection of the currently prevalent *grid-following* (GFL) WTs in the beginning of the restoration procedure can cause a recurrence of blackout as the grid is generally not stable enough (El-Zonkoly, 2015). The early participation of WPPs in successful bottom-up network energization can be facilitated instead by *grid-forming* (GFM) control of WTs, allowing them to operate together as an AC voltage source without relying on an external grid, and supply load in a power island.

GFM control of converters to integrate renewables at the distribution level, has been extensively researched upon for microgrids. However, only recently has it begun being applied to high-power applications. To the authors' knowledge, there are not enough in-depth studies addressing blackstart energization by high-power GFM renewable sources, connected at the transmission level. Since even at the distribution level, it is only recently—due to risks of uncontrolled islanding—that research has been conducted on microgrid islanding capabilities provided by GFM converters, for defence against blackouts in future power systems (Rocabert et al., 2012). Moreover, most existing microgrids are AC-systems, with DC-systems only now gaining momentum as they allow higher operational and control flexibility of the microgrid, enhancing its role in maintaining the reliability of future power grids (Arbab-Zavar et al., 2019).

## 1.2  Contribution

This study investigates the blackstart capability of a GFM OWPP, connected via HVDC—a challenging scenario due to not only faster energization transients but also more active components—and controlled with different grid-forming schemes, to compare their transient behaviour in such a challenging blackstart setup. The aim is to characterize the different techniques and compare their capability to deal with the energization transients in a controlled manner while maintaining stable voltage and frequency at the offshore terminal. There exist only a handful of such studies—on blackstart by HVDC/HVAC-connected GFM OWPPs—however, they choose one GFM method and focus on different aspects of the restoration process. In general, this paper aims at covering the lack of literature comparing the different grid-forming control strategies—typically developed for general purposes and driven by microgrid research-—in a specific and demanding task as wind power plant providing

blackstart services. While non-exhaustive, we think that this comparison can help direct future research in this area. To the authors' knowledge, such a study has not been done before.

In the next section of the paper, state-of-the-art on the role of wind energy in power system restoration has been reviewed. This is followed by an explanation of the general working principle of GFM control, along with a conceptual comparison of the four different control strategies considered for this study—on the energization of onshore load by an HVDC-connected GFM OWPP. Then the PSCAD model of the point-to-point HVDC-connected OWPP is described along with the different stages in the simulation of the energization sequence. Finally the transient behaviour of the different control techniques during the 80  various stages of energization are presented and discussed.

## 2   Wind energy for blackstart - Literature review

Large OWPPs can provide fast and fully-controlled, high-power, emission-free *green* blackstart services but there exists a gap between the present grid-code blackstart-requirements and current WT blackstart-capabilities as identified by Jain et al. (2019). Technological changes are neeeded to make WTs blackstart-*ready/able* and the technical challenges associated with 85  the different stages of energization of an HVDC-connected OWPP, along with control techniques to mitigate those issues have been discussed by Jain et al. (2018). A recent report by National Grid (2019b) also summarises the technological capability of non-traditional technologies like renewables and distributed energy sources to provide blackstart and restoration services. In the following, literature studies on WPPs—with different topologies—participating in power system restoration are presented, highlighting the associated research gaps.

### 90  2.1   WPP + Voltage source Hybrid

Traditional GFL WTs can be used with an external power supply (eg. diesel generator or energy storage) and a Synchronous Var Generator (SVG) or STATCOM, combining services into a joint/hybrid blackstart unit to facilitate WT participation in blackstart procedure as proposed in Aktarujjaman et al. (2006). The external supply provides startup power and sets the reference voltage and frequency for the isolated system, the SVG/STATCOM supports the Var requirement of the cables and 95  transformers and stabilizes the voltage, after which the WTs connect to meet the load power demand. Zhu et al. (2018) shows that earlier participation of WTs in the restoration procedure is feasible as GFM control allows blackstart and stand-alone island operation with better inherent synchronous-machine like inertial response during a transient, that can help absorb the initial impact of energization and ensure smooth load pickup, thus mitigating large voltage/frequency excursions that might occur during restoration. However, only the transients during load pickup and resynchronization to the grid have been studied, 100  while energization of collector lines, export cables and transformers, that present more challenges to transient stability during energization, are not shown. Additionally the major energization transients are dealt by the energy storage system and SVG, while the WTs behave only as passive GFL power-sources to meet the load-demand during the last stages of restoration.

## 2.2 HVAC-connected WPP

Recent studies by Martínez-Turégano et al. (2018) and Aten et al. (2019) demonstrate the potential capability of HVAC-connected OWPPs to blackstart onshore grid using GFM controls in less than 25% WTs and assuming adequate wind resource. The results show that it is possible to do sequential energization of the array-cables and WT transformers, starting with one WT energizing its string followed by others synchronizing to it and then sharing the control of voltage and frequency. Shorter cable sections are energized first until enough WTs are connected to absorb the Var generated by subsequent cable sections. However, according to Elia (2018) and National Grid (2019b), a large gap to bridge is the energization of the export link while meeting grid code requirements.

## 2.3 HVDC-connected WPP

HVDC with Voltage Source Converters (VSC) can also be used as a standby facility for blackstart and restoration of the onshore AC grid, as demonstrated by the excellent voltage and frequency control performance in real system tests done by Jiang-Hafner et al. (2008), proving for the first time that VSC-HVDC helps reduce restoration time while facilitating a safer and smoother restoration process with lower investment and maintenance cost. With HVDC transmission gaining momentum as the preferred choice for longer distance connections to larger OWPPs, Sørensen et al. (2019) shows that the Skagerrak-4 VSC-HVDC link between Norway and Denmark (DK) can be successfully used to ramp-up the voltage of an islanded 400/150 kV DK-network to energize overhead transmission lines, transformers and block load, followed by synchronization to continental EU. Additionally, a top-down restoration test of the NEMO link between Belgium and the UK also demonstrates the capability of the VSC-HVDC interconnector to energize a dead Belgian grid from the live UK side (Schyvens, 2019).

Simulation results by Becker et al. (2017) show, although without any details of the transformer/cable energization transients, that a VSC-HVDC connected OWPP can respond to onshore load changes and participate in load restoration. Cai et al. (2017) analyzes the inrush current of transformers and cables (HVAC/HVDC) using electro-magnetic transient simulations, but with a diesel generator to pre-charge the offshore converter that then energizes the offshore collector grid, and the onshore converter pre-charged by the onshore AC-grid, contrary to what is expected from an OWPP to provide blackstart service. Simulation results presented by Sakamuri et al. (2019) demonstrate, for the first time, an HVDC-connected OWPP with GFM control, sequentially energizing the offshore AC network including transformer, cables and converter through a pre-insertion resistor, followed by HVDC link energization and onshore converter pre-charging and de-blocking for picking up block load, successfully participating in restoration as a blackstart unit. However, the energy imbalance in the HVDC link during the DC-side uncontrolled pre-charging of the onshore converter leads to a significant dip in HVDC voltage and large transients in the offshore and onshore converter cell voltages and valve currents.

In addition to enabling blackstart and islanding capabilities of WTs, GFM control can also allow the use of Hybrid-HVDC connection with a diode rectifier unit (DRU) instead of the offshore VSC. The application of controls proposed in Blasco-Gimenez et al. (2010) for an OWPP to ramp-up the offshore AC grid voltage and control frequency, considering it as an inverter-based microgrid, has shown improved steady-state regulation during islanding when the DRU-HVDC is not conducting, and

smooth transition to current-control mode during grid-connected operation. This significantly reduces the cost-vs-performance, due to lower losses (especially for higher power levels) and lesser capital cost, along with increasing efficiency and reliability due to a lower probability of commutation failure than a VSC (Andersen and Xu, 2004).

## 3 Grid Forming Control

The current turbine and converter controls are designed assuming a strong grid connection point which means that the grid-side converter of the WT latches onto a pre-existing voltage signal provided by the onshore grid in case of an AC-connected OWPP, or produced by the offshore HVDC converter operating in voltage-frequency control mode in case of HVDC-connected OWPP (Bahrman and Bjorklund, 2014). However, to allow outward-energization of the network of inter-array cables and transformers, create a power island that can supply local loads and energize the HVDC link converters and export cable with the ultimate aim

to supply onshore block load, the WT should be able to produce its own voltage signal. This requires GFM control, traditionally referred to as *voltage-injecting* control, as opposed to the conventional GFL or *current-injecting* control. The two control philosophies are very well explained by (Rocabert et al., 2012). GFM WTs can also minimize the use of diesel generators that are currently employed offshore to supply backup auxiliary power required for energization. Although most modern WTs have an on-board UPS to power communications, protection and control for a few hours during emergency shutdown (Göksu

et al., 2017), a larger internal backup supply may be required for self-starting the WT for blackstart, especially after extended shutdown periods.

GFM control of PECs has been well studied for microgrids, where the role of the converter is to act as an interface between the small-scale distributed/renewable power generation units and the consumption points, leading to inertial decoupling of the rotating machines and making the microgrid system susceptible to oscillations caused by network disturbances. GFM allows a

155 PEC to mimic synchronous generators for droop-based load-sharing, synthetic-inertia emulation, synchronized and stand-alone operation and blackstart behaviour, ensuring voltage and frequency stablility in low-inertia microgrids during varying loads, network disturbances and system configurational changes e.g. islanding $\Longleftrightarrow$ grid-connected (Tayyebi et al., 2018).

An OWPP is like a microgrid rich in power electronics, although very different in that the voltage and power levels are much higher. Moreover, OWPP operators maintain a large amount (>100s) of WT-assets that are located very far from each

160 other. Current sharing techniques for low rated inverters like the centralized controllers and the master-slave approach can be used only for paralleled systems that are close to each other and interconnected through high-bandwidth communication channels (Rocabert et al., 2012). These communication-based solutions cannot be used for microgrids spread across several kilometres, as ensuring globally available, bidirectional, reliable, robust, low-power and secure communication architecture becomes increasingly costly. Moreover, longer links increase delays which is undesirable in cases where a fast (high-bandwidth)

communication is required. This gave way to *droop control* algorithms with a hierarchical structure being used in microgrids, especially for islanded operation of many micro-sources located far away from each other (Pogaku et al., 2007). Although rated at much lower power, these GFM droop-based strategies can be extended to high power WTs for stable distributed op-

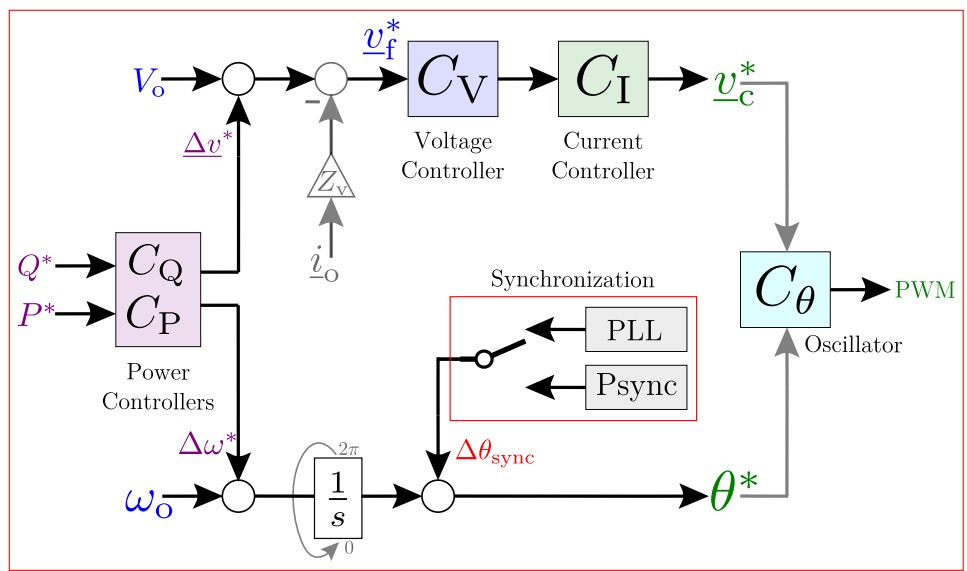

**Figure 1.** Grid-forming control structure consisting of current control loop $C_I$, the voltage controller $C_V$ and outer real/reactive power control loops $C_{P,Q}$.

eration in islanded mode, at variable loads and wind speeds, as demonstrated by Kanellos and Hatziargyriou (2008) and also Blasco-Gimenez et al. (2010).

### 3.1  Control Structure

According to the definition in Rocabert et al. (2012), GFM converters are controlled in closed loop to work as ideal AC voltage sources (low-output impedance), while GFL converters are controlled as current sources with high parallel output impedance and can't operate in islanding/stand-alone mode as they require a GFM converter or local synchronous generator to set the bus voltage and frequency.

The structure of GFM control consists of different functional blocks, as shown in Fig. 1. The main objective of GFM control is to operate the VSC as an ideal AC voltage source of given amplitude $V_0$ and frequency $\omega_0$. This requires most importantly, a *voltage control* loop $C_V$. The short-comings of the single-loop approach, explained in Zeni et al. (2015), are already known from switch-mode power supplies and electrical machine drives as over-currents during transients and faults cannot be limited due to the lack of an explicit closed-loop current controller. Additionally sensitivity to disturbances and plant-parameter fluctuations eliminates open-loop control as a good choice. The most commonly used alternative thus, is the nested/cascaded voltage-current controller, in which a faster inner *current control* loop $C_I$ is added (Zeni et al., 2015). $C_I$ is designed to have a relatively smaller time constant than $C_V$ for decoupling the control loops. The controllers are in the synchronous reference frame that uses an angle $\theta^*$ (for $abc \rightleftharpoons dq$ transformation) obtained from the *synchronization block* (Green and Prodanović, 2007).

While grid-feeding converters require perfect synchronism with the voltage at the point of connection to accurately regulate the power exchange with the grid, in the case of GFM converters the synchronization system must provide precise signals for both islanded and grid-connected modes of operation. It works as a fixed frequency oscillator in the former case, while slowly varies the phase-angle and frequency of the island voltage during the reconnection transient to resynchronize with the grid voltage, in the latter. The most extended method used in grid-connected operation is a Phase Locked Loop (PLL), also called *voltage-based* synchronization as the frequency and phase-angle of the grid voltage vector is used for control. However, enhancements are needed to ensure stability under unbalanced and distorted voltage conditions as voltage sag, weak grids or off-grid operation can lead to instabilities. Alternatively *power-based* synchronization can also be used as the structure of the swing-equation that governs synchronous machine dynamics, can be equated to that of a PLL, in the sense that the PLL structure can be modified to extract the derivative term of the frequency ($\sim$inertia) and the speed variation ($\sim$damping), as shown in van Wesenbeeck et al. (2009). This presents a more stable solution and allows the power controller to also act as the synchronization block.

The outer *power* control loops $C_{\mathrm{P,Q}}$ are required to regulate the real ($P$) and reactive ($Q$) powers exchanged with the grid (in grid-connected mode) or meet the demand set by the load (in islanded mode), while ensuring communication-less power sharing between the multiple paralleled inverters. The simplest method for this, by only relying on local measurements, is the *droop control* scheme, which was initially introduced for synchronous generators in utility scale grids, and now is well incorporated into microgrids (Arbab-Zavar et al., 2019). The primary level of the *3-level hierarchical* control, explained in Guerrero et al. (2011), employs droop control equations, based on interconnecting impedance X/R ratio, to mimic the self-regulation capability of a grid-connected synchronous generators and allow power sharing in microgrids without using critical communication links (Rocabert et al., 2012).

Although easy to implement with high reliability and flexibility, traditional droop control suffers from an inherent tradeoff between load-sharing and voltage-regulation, load dependent frequency-deviation, slow dynamic response due to filters for power measurement, and non-linear load-sharing issues due to harmonics. A variable *virtual impedance* $Z_{\mathrm{V}}$ can be used to add harmonic droop characteristics with additional damping and improve tradeoff between current harmonic sharing and voltage total harmonic distortion, by adjusting the output impedance seen at different frequencies. Additionally this allows intelligent mode-switching with soft-start to take advantage of the fast converter response while avoiding large transients (Guerrero et al., 2011). In the last decade, several different GFM control schemes have been proposed in literature, of which four have been chosen for this study and explained in the next section.

## 3.2 Control Strategies

The traditional droop based power controllers can be replaced with more complex controls to replicate the system-level functionalities of synchronous generators like inertia and damping characteristics, frequency/voltage droop, self-organising parallel operation and automatic power sharing. The VIrtual Synchronous MAchine (VISMA) concept, introduced by Beck and Hesse (2007), uses power-based synchronization with a detailed implementation of the electro-mechanical model of a synchronous machine in its power control loop. This eliminates the need for PLL and allows conventional and proven grid operation with

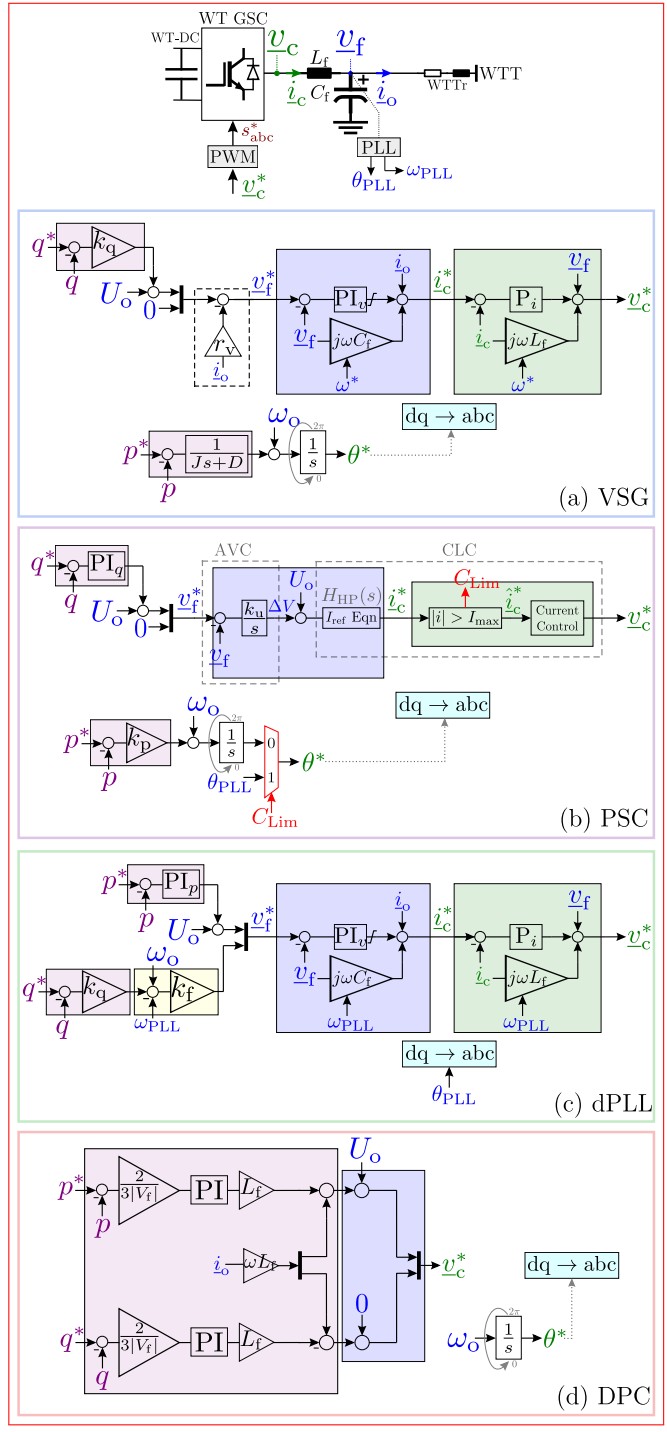

**Figure 2.** Control structure for (a) Virtual Synchronous Generator (VSG), (b) Power Synchronous Control (PSC), (c) Distributed-PLL based (dPLL) control and (d) Direct Power Control (DPC) GFM strategies. The coloured boxes match the different blocks with their function (blue→voltage control, green→current control and purple→power control) in the general GFM control structure, given in Fig. 1.

the usual static and dynamic properties that are characteristic to synchronous generators (both desired and undesired). Different detail levels of the VISMA implementation are listed in D'Arco and Suul (2013) while Lu and Cutululis (2019) gives a review of different control methods that mimic the operation of a rotating synchronous machine, for example Synchronverter (Zhong and Weiss, 2009) and PLL-based swing-equation emulation (van Wesenbeeck et al., 2009). Moreover, improvements have been made to the VISMA concept for example recently, non-linear control based GFM strategies relying on the duality between PECs and synchronous machines have been proposed. This includes Machine-Matching (Arghir et al., 2018) and Virtual Oscillator Control (Johnson et al., 2017), that provide steady-state droop-like behaviour with a faster and better damped response during transients.

### 3.2.1 Virtual Synchronous Generator (VSG)

The virtual synchronous machine concept implemented here, shown in Fig. 2(a), is based on D'Arco et al. (2015a). It uses standard cascaded voltage-current control in the synchronous rotating $dq$ reference frame for voltage control and current limitation. The behaviour of a synchronous machine is mimicked by using the swing equation—see Eq. (1)—for power control. This also helps in power-synchronization by generating the frequency reference and synchronization angle to control real power exchange with the grid, similar to a synchronous generator. The reactive power controller is based on standard inductive line $Q - V$ droop, that adjusts the voltage amplitude reference for controlling the reactive power exchange with the grid. A virtual resistance is added to reduce sensitivity to small grid disturbances by providing additional damping and reduce the synchronous oscillations of droop controlled converters (Sun et al., 2019). The swing equation controller, essentially a low pass filter, can be replaced with e.g. PID/Lead-Lag controllers, for enhanced electro-mechanical dynamics, adjustable characteristics like independent tuning of inertia, damping and steady-state droop, or high non-linear behaviour during grid faults and connection/disconnection processes (Sun et al., 2019).

### 3.2.2 Power Synchronous Control (PSC)

The power-synchronization law presented in VSG above uses the swing equation where the power difference drives the rotor speed dynamics which is then integrated to get the electrical angle i.e. double integration for $P$-$\theta$ transfer function. This can be simplified using the PSC control structure explained in Zhang (2010), as shown in Fig. 2(b). Here the phase angle is directly obtained by a single integration of the power difference, as given in Eq. (2). Due to one less integrator, PSC has higher stability margin. However, no virtual inertia or damping is present due to absence of rotor dynamics.

$$P_{\mathrm{m}} - P_{\mathrm{e}} = J\omega_0 \frac{\mathrm{d}\Delta\omega}{\mathrm{d}t} + D\omega_0\Delta\omega, \ \frac{\mathrm{d}\Delta\theta}{\mathrm{d}t} = \Delta\omega \Longleftarrow \text{ Swing Equation} \tag{1}$$

$$\frac{\mathrm{d}\Delta\theta}{\mathrm{d}t} = k_{\mathrm{p}}(P_{\mathrm{ref}} - P) \qquad\qquad \Longleftarrow \text{Power Synchronization Law} \tag{2}$$

The voltage control is governed by Eq. (3), where an AC Voltage Controller (AVC) is used similarly to the exciter of a synchronous machine, except with integral control instead of the typical proportional control, as shown in Fig. 2(b), to supress high-frequency disturbances. Active damping of the grid-frequency resonant poles is additionally implemented using a high-

 pass filter $H_{\mathrm{HP}}(s)$, described by Eq. (4), in the Current Reference generating block, given by Eq. (5).

$$\underline{v}_{\mathrm{C}}^* = (V_0 + \Delta V) - H_{\mathrm{HP}}(s)\underline{i}_{\mathrm{C}} \qquad \Longleftarrow \text{Voltage Control} \tag{3}$$

$$H_{\mathrm{HP}}(s) = \frac{k_{\mathrm{v}}s}{s + \alpha_{\mathrm{v}}} \qquad \Longleftarrow \text{High Pass Filter} \tag{4}$$

$$\underline{i}_{\mathrm{C}}^* = \frac{1}{\alpha L_{\mathrm{f}}}[(V_0 + \Delta V) - \underline{v}_{\mathrm{F}} - j\omega_0 L_{\mathrm{f}}\underline{i}_{\mathrm{C}} - H_{\mathrm{HP}}(s)\underline{i}_{\mathrm{C}}] + \underline{i}_{\mathrm{C}} \Longleftarrow \text{Current Reference Equation} \tag{5}$$

$$\underline{v}_{\mathrm{C}}^* = \alpha L_{\mathrm{f}}(\underline{i}_{\mathrm{C}}^* - \underline{i}_{\mathrm{C}}) + j\omega_0 L_{\mathrm{f}}\underline{i}_{\mathrm{C}} + \underline{v}_{\mathrm{F}}, \, \alpha = \mathrm{BW} \qquad \Longleftarrow \text{Current Control} \tag{6}$$

Although not included in this study, the PSC uses the current reference generated above—as it gives an indication of the actual current—for over-current limitation in the Current Limitation Controller (CLC). A standard $dq$ Current Controller, tuned for a set bandwidth of $\alpha$ rad/s, as given by Eq. (6), is used. In *fault* mode, the CLC limits the current output of the converter to $I_{\max}$ and generates a selector signal $C_{\mathrm{Lim}}$, to disable the power-synchronization and switch to conventional PLL-based synchronization. However, in *normal* mode ($|I| < I_{\max}$), Eqs. (5) and (6) simplify to Eq. (3), for voltage control as described above. The PSC has demonstrated strong performance in weak networks.

### 3.2.3 Distributed PLL-based (dPLL) control

Contrary to power-synchronization implemented in VSG and PSC, the dPLL control structure, based on Yu et al. (2018) and shown in Fig. 2(c), uses voltage-based synchronization by using a PLL for frequency control. Originally developed for DRU-connected OWPPs, the real power controller is used to generate the $d$-axis voltage reference as power flow is determined by offshore voltage, and a droop controller regulates frequency to share the DRU reactive power demand. Instead of the conventional approach of setting the $q$-axis voltage reference to 0, since the PLL output can be used as an indication of frequency deviation, a Frequency Control loop characterized by Eq. (7), is embedded in the $q$-axis.

$$v_{\mathrm{fq}}^* = k_{\mathrm{f}}(f^* - f) \tag{7}$$

Yu et al. (2018) demonstrates frequency controllability with plug-and-play capability providing successful sequential start-up of the GFM WTs and automatic synchronization of the offline WTs during connection with minimal impact, to supply the Var required to energize transformers, filters and finally ramp-up the offshore voltage and start delivering active power to the onshore grid. However, only the start-up and synchronization of an islanded OWPP to an energized onshore synchronous power system via a DRU-HVDC link is studied while the energization of export cable and onshore converter, expected from a blackstart service provider, was not looked into.

### 3.2.4 Direct Power Control (DPC)

Lastly, a control scheme based on direct power control, originally introduced by Noguchi et al. (1998) has been implemented. In DPC, the instantaneous powers are controlled without requiring AC voltage sensors, PLL or an inner current controller, by using a look-up table and hysteresis comparators on the power errors to select the optimum switching state of the converter.

Since then it has undergone many enhancements to deliver improved performance like using space vector modulation for constant switching frequency, employing sliding mode control for robustness, and model predictive control for the multivariable case. The implementation used in this study is based on an improved DPC described in Gui et al. (2019), in which grid-voltage modulation allows linearization of the original non-linear system resulting in ease of control design and good transient-response and steady-state performance. The control structure, as shown in Fig. 2(d), has been derived in Gui et al. (2019) using the *instantaneous pq theory* (Akagi et al., 2017) and results in a standard VSC *dq* current control structure without the need for a PLL. In the GFM control implementation for this study, the voltage reference is given in place of the grid voltage magnitude and a virtual phase angle is used in place of PLL generated voltage phase angle, based on Cheng and Nian (2016).

The block schemes of the four GFM control strategies explained above viz. VSG, PSC, dPLL and DPC, are shown in Fig. 2. In the figure, vectors are denoted by $\underline{x} = x_{\mathrm{d}} + jx_{\mathrm{q}}$, while scalars by $X$. The controls are implemented in pu. The coloured boxes in Fig. 2 show how the different blocks of each control scheme fit functionally—blue for voltage control, green for current control and purple for power control—into the general control structure presented in Fig. 1 and explained in section 3.1.

## 4 Model Description

A model of the system schematic shown in Fig. 3 and based on Sakamuri et al. (2019) has been developed in PSCAD. It consists of a 400 MW GFM OWPP connected to the onshore AC grid by means of a 200 km long 1200 MW $\pm 320$ kV symmetrical monopole point-to-point HVDC link, as shown in Fig. 3(a).

A Detailed Equivalent Model has been used for the Half-Bridge Modular Multilevel Converters (MMC) of both terminals of the VSC-HVDC link. This represents each sub-module as an equivalent circuit model for simplification while solving the network, and then converting back to the sub-modules, thus giving a fast solution along with information about what happens inside the sub-modules. The offshore terminal (T2) MMC is controlled in GFL mode since the offshore AC network voltage is formed by the GFM OWPP, so the converter regulates the HVDC link voltage $V_{\mathrm{DC}}$ and reactive power injection $Q_2$ into T2. At the onshore terminal (T1), the MMC is controlled in GFM mode to regulate the onshore AC voltage magnitude $V_1$ and frequency $f_1$, in the scope of the blackstart case study performed in this paper. The MMC models used have standard MMC inner control loops, such as cell voltage balancing and circulating current suppression, and the control structure for the $V_{\mathrm{DC}}$-$Q_2$ (for T2) and $V_1$-$f_1$ (for T1) modes, can be seen in Sakamuri et al. (2019). Frequency dependent (phase) models of PSCAD are used for the HVDC export cable. The HVDC converter transformers models include magnetic characteristics such as saturation and inrush current. Finally a Pre-Insertion Resistor (PIR) that is bypassed after a Pre-Insertion Time (PIT) by using coordinated Main-Auxiliary Breakers (MB-AB), is used for limiting the transient magnetic-inrush current peak during hard-switching energization of the HVDC transformer.

The OWPP consists of 50 Type-4 (fully-rated PEC interface) 8 MW WTs, as a partially-aggregated model shown in Fig. 3(b), based on Muljadi et al. (2008). It consists of 9 individual $\mathrm{WT}_{1-9}$ models on the first string, the second string with $\mathrm{WT}_{10-18}$

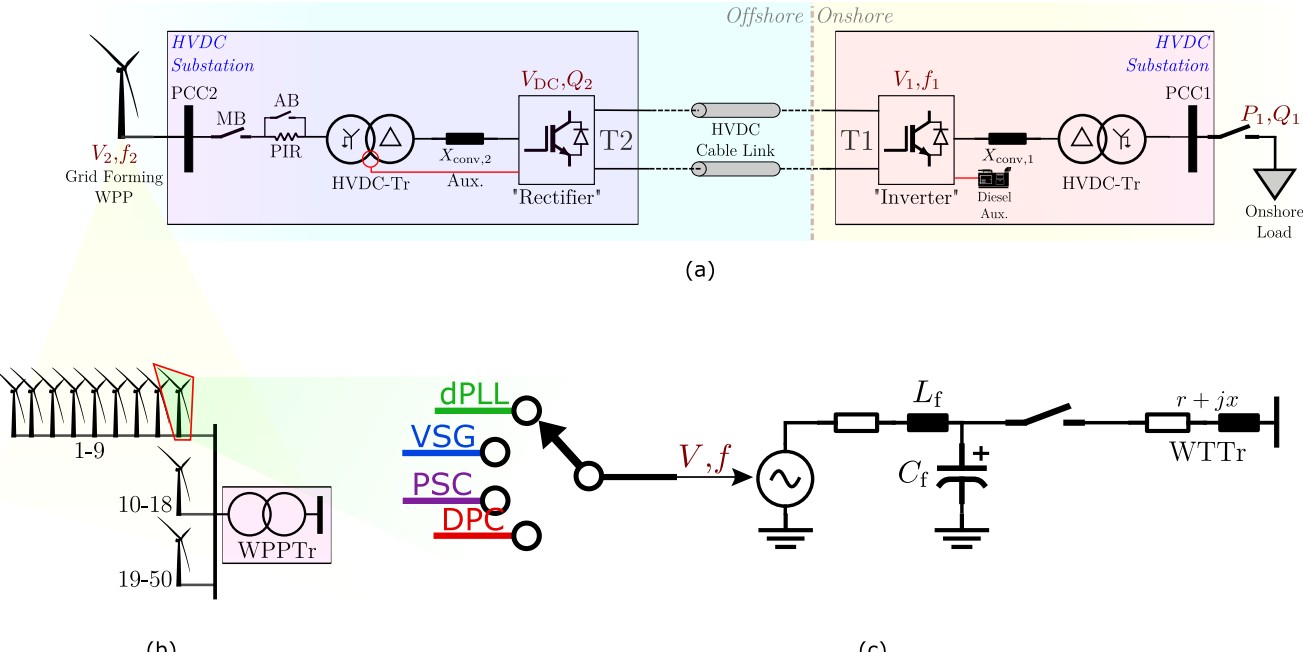

**Figure 3.** Schematic of the implemented PSCAD model of the system under study. This figure shows (a) the 2-Terminal HVDC link, with GFL offshore-MMC and GFM onshore-MMC, (b) the partial aggregation used to model the OWPP, and (c) the average model of the GFM WT-GSC implemented with 4 different control strategies operating in islanded mode.

aggregated into a 72 MW WT model, and the remaining 32 $WT_{19-50}$ aggregated into one 256 MW model. Coupled $\pi$-section models are used for the 66 kV array cables. Lastly the WT is modelled as a GFM unit operating in islanded mode, and so the Grid-Side Converter (GSC) is modelled as a voltage source (average model) controlled by the four different GFM strategies, viz. VSG, PSC, dPLL and DPC, that are explained in section 3.2. This is shown in Fig. 3(c).

## 4.1 Assumptions

Several simplifications have been made, mainly removing modelling details deemed not relevant for this study.

Firstly, the WT Rotor-Side Converter (RSC) and changes to the turbine controller that are required for GFM operation, have not been modelled. In conventional GFL operation of the WT, the RSC is controlled to extract maximum power from the generator while the GSC maintains power balance to control the DC link voltage of the back-to-back PEC interface of the WT and the reactive power output at the AC terminal. However, in GFM mode, the GSC cannot control the WT-DC link and reactive power anymore with the required generator torque and real power being set by the AC-load, not the turbine controller, which now has to regulate the speed using pitch control (and especially avoid overspeeding during low AC-load and high winds). Hence the RSC control requires changes to be able to maintain the DC link voltage constant by ensuring real power balance (Pérez et al., 2019). Since the WT rotor and DC-link dynamics are outside the scope of this study, the model assumes

a constant WT-DC link voltage. Additionally an average voltage source model is used for the GSC with focus on dynamics not faster than the bandwidth of the inner current control loop. Moreover, the WT transformer is modelled as a pure electrical impedance $r + jx$ without any magnetic characteristics as it can be soft-started along with the WT voltage ramp-up, to avoid magnetic inrush and saturation effects.

Secondly, for this study, although power sharing between the WTs inside the WPP is controlled by including the outer power control loops, the WTs are started-up *simultaneously* as opposed to a more realistic *sequential* energization e.g. in Yu et al. (2018), as the study mainly focuses on the capabilities of the GFM OWPP as a whole, to provide blackstart services to the onshore grid while dealing with offshore network transients due to energization of the large converter transformer, HVDC converters and export cable—in a controlled manner. This puts any synchronization dynamics of multiple GFM PEC-interfaced WTs out of the scope of this study.

## 4.2 Controller Tuning

In this section, the tuning criteria for the control parameters of the different control schemes are presented. In a cascaded control structure, the inner loops are designed to achieve a fast response while the outer loops are tuned for regulation and stability.

Assuming a switching frequency of the wind turbine converter of 1 kHz, and with the main objective of the current controller to have a fast response, a bandwidth ($\alpha$) of 200 Hz has been selected (Yazdani and Iravani, 2010). A proportional controller has been used as any steady state error can be taken care of by the outer voltage controller. This results in the same value ($\alpha L_\mathrm{f}$) of the proportional gain for VSG and dPLL, as that already used in the current control Eq. (6) for the PSC scheme. The voltage controller on the other hand is tuned to provide zero steady-state error with a bandwidth of 40 Hz and a phase margin of $45°$ (Yazdani and Iravani, 2010).

The outer power control loops are tuned to be sufficiently slower than the voltage and current controls to avoid coupling between the control levels. Since the WPP is operating in islanded mode, there is no grid to exchange power with based on a set reference, rather the real and reactive power demands are set by the load. For the DPC, the power control loops take the form of standard vector current control, as shown in Gui et al. (2019). Since $p \propto i_\mathrm{d}$ and $q \propto i_\mathrm{q}$, tuning the DPC power controller is equivalent to tuning a slower current controller—a natural frequency of 4 Hz with damping ratio of 0.74 has been chosen.

A unique control loop in the dPLL scheme is the Frequency Control loop that includes the cascaded voltage-current controller dynamics along with PLL, as shown in Yu et al. (2018). This requires tuning of the PLL and the controller gain $k_\mathrm{f}$ of Eq. (7). The PLL has been tuned to be critically damped with a bandwidth of 0.5 Hz—slow enough for the slow voltage controller which significantly reduces damping ($\sim$ phase margin) in the system moving it closer to instability. While increasing $k_\mathrm{f}$ leads to a faster response but with reduced damping (Yu et al., 2018), a low value was found to result in oscillations in the shared powers. As a trade-off, $k_\mathrm{f}$ has been tuned for the frequency control loop to have a bandwidth of 0.5 Hz. Overall, these values ensure a high phase margin ($82°$) for stable operation.

**Table 1.** Simulation events of the energization sequence.

| Stage | Time [s] | Events |
|---|---|---|
| 1 | 0 | WTs energized simultaneously and operate in GFM mode. |
| 2 | 1.3 | GFM WPP is connected to energize PCC-2 and |
|   |   | MB is closed to insert PIR for energizing the offshore HVDC transformer and pre-charging the offshore MMC cells. |
|   | 1.6 | PIR is bypassed after PIT (= 0.3s) by closing AB. |
| 3 | 2.1 | Offshore MMC is de-blocked to control the HVDC voltage—HVDC link is energized |
| 4 | 2.5 | (a) Controlled pre-charging of onshore MMC's upper arm cells with lower arm bypassed. |
|   | 2.8 | (b) Controlled pre-charging of onshore MMC's lower arm cells with upper arm bypassed. |
|   | 3.1 | (c) Controlled pre-charging of onshore MMC finished; both arms blocked. |
| 5 | 3.3 | Onshore MMC is de-blocked to control voltage and frequency—Onshore AC PCC-1 is energized. |
| 6 | 4 | Onshore 30 MW block load is connected. |

Similar to the dPLL, the cascaded voltage-current controller of the VSG moves the system closer to instability at low bandwidths, due to a significant reduction in system damping ($\sim$ phase margin). This is justified by D'Arco et al. (2013,2015b), which show that certain eigen values close to the imaginary axis are sensitive to the proportional gain of the voltage controller $k_{pv}$, with higher values improving the stability of the system. Moreover, lower switching frequencies seem to shift the root locus closer to imaginary axis restricting the range of stable operating points. Sun et al. (2019) shows the complexity associated with tuning VSG for damping the different oscillations, implicitly caused due to coupling terms in the state-space matrix. While virtual resistance helps damp the intrinsic synchronous mode of a single droop-controlled VSC, it introduces coupling between active and reactive powers, especially for multiple paralleled VSCs. The inertia term $J$ can then be tuned to provide further attenuation, but this introduces sub-synchronous oscillations which deteriorate with increased inertia. Finally according to Sun et al. (2019), since smaller droop gains reduce the inter-oscillations, values for $D$ and $k_q$ from D'Arco et al. (2015b) have been taken as initial estimates and then fine tuned for stable operation. It is important to note that improving one oscillation mode can trigger another, and that the tuning strategy used for a single VSG might not be applicable to multiple VSGs (Sun et al., 2019). Consequently the VSG control strategy implemented here—with swing equation—has been found to be limited in its damping of oscillations. With a slow voltage controller—smaller $k_{pv}$—the sub-synchronous mode moves closer to origin, while the synchronous mode improves in damping (D'Arco et al., 2015b). So the voltage controller has been set to as low as possible a bandwidth ($\sim$ 150 Hz) while ensuring good damping and stable operation, with appropriately tuned outer loops.

The complete set of tuned parameter values are listed in Table A2.

## 5  Simulation Results

In this section, the results of the dynamic simulations performed in PSCAD are presented. The energization sequence, events of which are described in detail in Table 1, is based on Sakamuri et al. (2019), but includes an extra stage of *DC-side controlled*

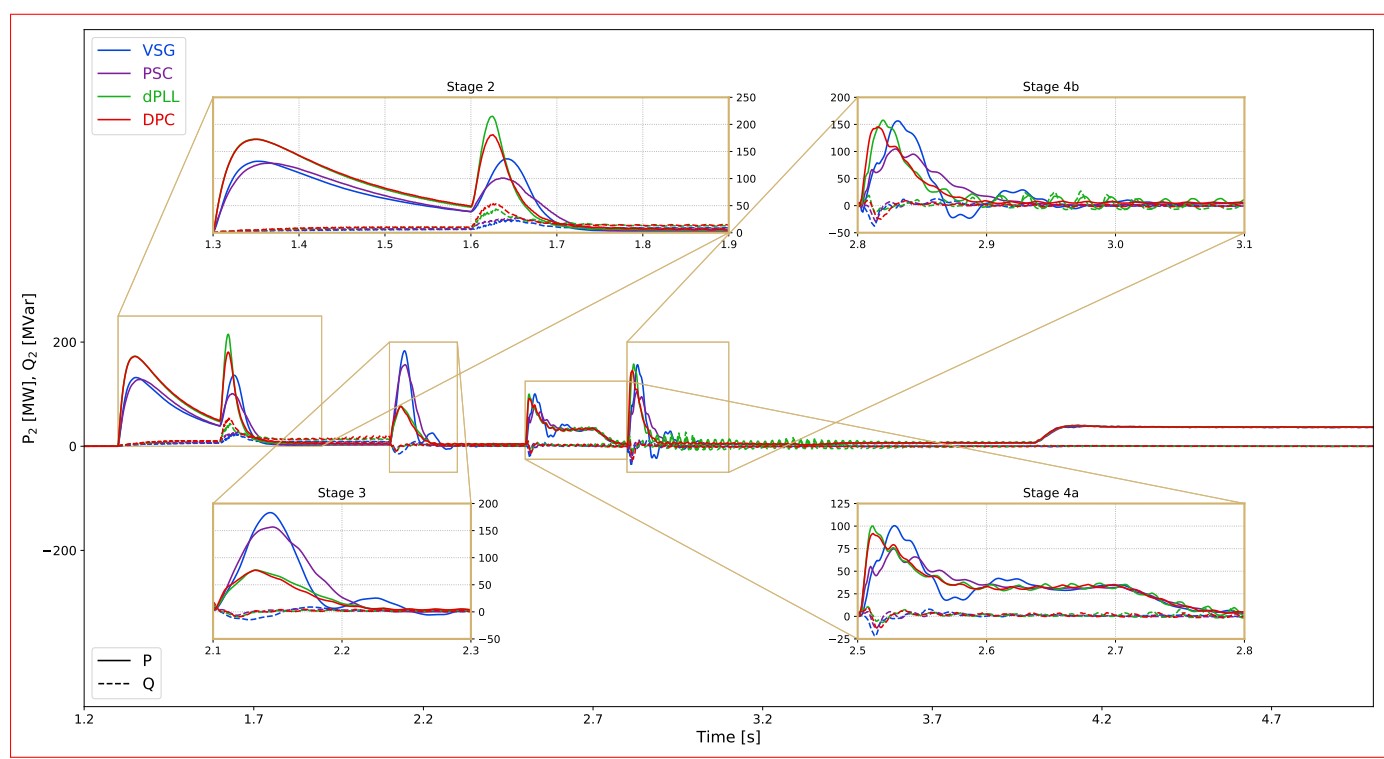

**Figure 4.** Real (solid line) and reactive (dotted line) power output of the offshore WPP with zoomed insets to show transients in selected stages of the energization sequence.

*pre-charging* of the onshore MMC cells along with the outer power control loops enabled for real and reactive power sharing amongst the WTs inside the WPP. The entire sequence is simulated, however, the main focus is on testing the characteristics of the different control strategies in enabling the OWPP to deal with the energization transients—so we focus on the real and reactive power outputs of the WPP, and the voltage and frequency at the offshore PCC-2. Hard-switching is used here despite the advantages of soft-start energization, as the former is more demanding on the GFM OWPP in terms of the transients linked to energization of transformers, cables and HVDC link.

Figure 4 shows the waveforms for the real and reactive power outputs of the WPP during the different stages of the energization sequence. Since the GFM OWPP is operating in islanded mode, the real and reactive power demand is set by the load, which depends on the particular stage of energization. For Stage 2, it is the reactive power required for magnetic energization of the offshore HVDC transformer and *AC-side* pre-charging of the offshore MMC cells. A PIR is inserted for PIT duration to limit the inrush peak. In Stage 3, power is required to energize the HVDC cable when the offshore MMC is de-blocked to control the HVDC link voltage, while in Stage 4, the *DC-side* pre-charging of the onshore MMC cells draws power from the OWPP to maintain the energy balance on the HVDC link. Finally the OWPP supplies power to match the onshore block load in Stage 6. It is clear from the $PQ$ waveforms shown in Fig. 4 that there are some differences in the transient behaviour of the

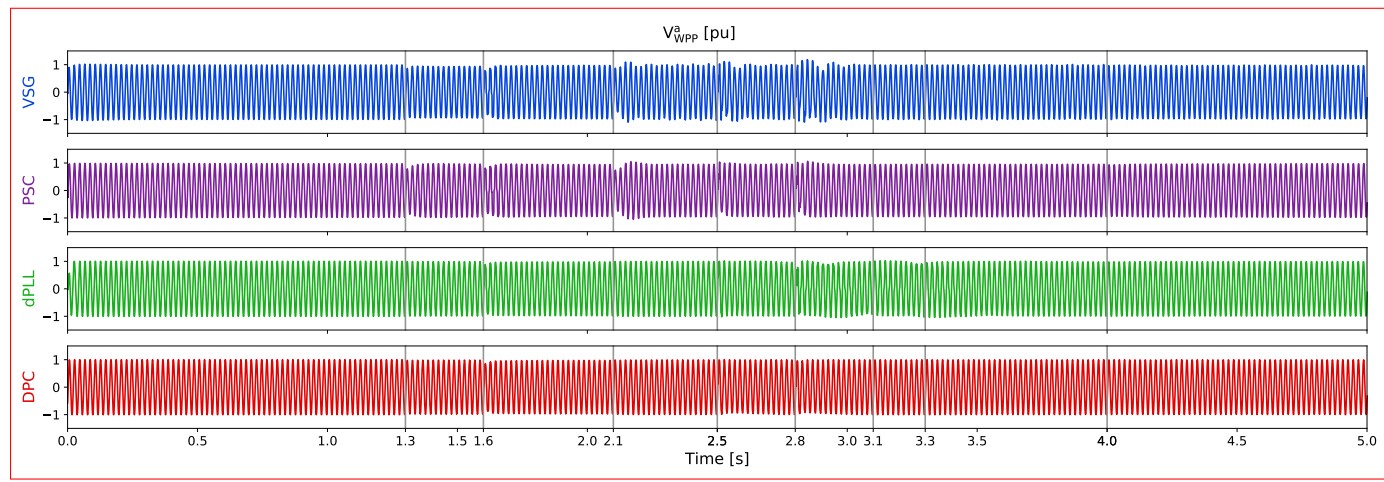

**Figure 5.** WPP output phase voltage (in pu) for the different GFM control schemes.

four control strategies, despite having an overall similar profile. PSC and VSG show a delayed response, as indicated by the delayed peaks, in stages 2 (at 1.6s), 3, 4a and 4b. This is due to the right half plane zero in the closed loop transfer function $\frac{\Delta P}{\Delta \theta}(s)$ for the power-based synchronization methods, as shown in Zhang et al. (2010). The dPLL shows a synchronous and

sub-synchronous mode getting excited at Stage 4b probably due to a change in system coupling. However this is damped after some time due to the current control. Finally the DPC, due to its overall standard current control structure, shows good damping of oscillations, as apart from over-current limitation, the current controller's function is also to damp resonance modes (Zhang, 2010).

Since the scope of this study is to focus on the OWPP behaviour as an AC voltage source during the different stages of

energization, the waveforms for the voltage and frequency at the offshore PCC-2 are presented in Figs. 5 and 6, respectively. The GFM WPP, controlled as an AC voltage source, has different characteristics based on the control method used. The *V* waveform in Fig. 5 show that the OWPP with the four different GFM controls can successfully energize the transformer, cables and MMC cells and supply the onshore load, while maintaining a stable voltage at the offshore PCC-2, with the transient distortions during the different stages being recovered fast by the GFM controls.

An interesting observation in Fig. 5 is that during Stage 4, there is oscillation in the voltage for VSG, which transiently increases as the active power increases (Fig. 4) with frequency decreasing at 2.5/2.8 s (Fig. 6). This is linked to the energy imbalance in the HVDC link as the offshore MMC cells discharge during the charging of onshore MMC cells, to which the WPP then reacts by producing the required active power and absorbing the reactive power generated from the capacitor charging, as shown by the negative Var curves in Fig. 4—Stages 4a and 4b. Comparing with synchronous generators, this can

be understood similar to transient rotor-angle instability during large load changes in weak networks, which in our case is due to virtual resistance and a transient change in network coupling during capacitor charging. To enhance stability of the VSG, a virtual Automatic Voltage Regulator (AVR) can be used like in synchronous generators and is the work of future studies.

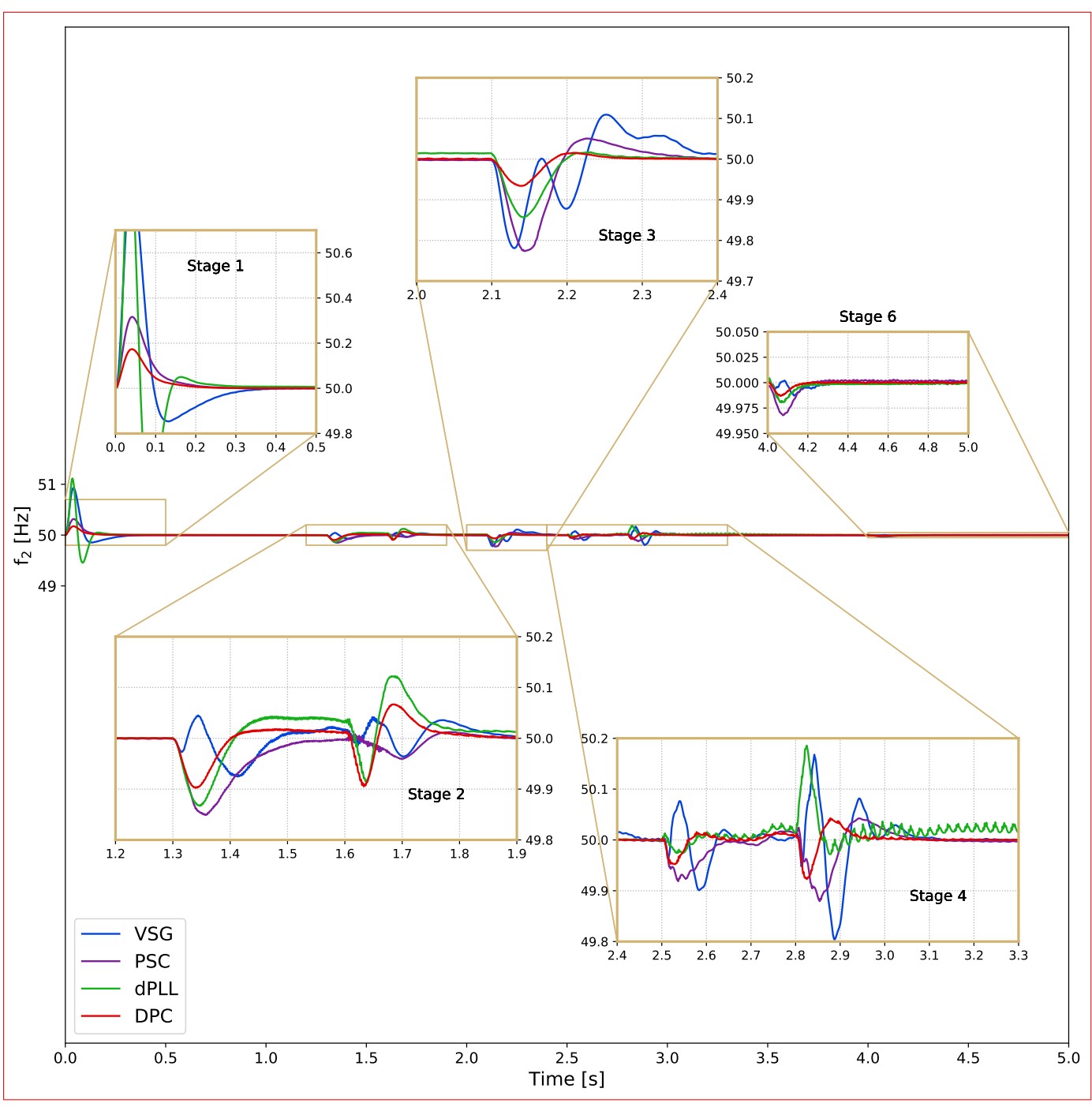

**Figure 6.** Frequency at offshore PCC-2 with zoomed insets to show transients in the different stages of the energization sequence.

Having looked at voltage, the other key aspect of the grid-forming WPP is its frequency response. The $f$ waveform in Fig. 6 shows significant differences in the frequency transients highlighting the frequency control characteristics of the different grid-forming methods, although overall the frequency swing is in the range of 49.8–50.2 Hz, except in Stage 1.

At startup (Stage 1), there is a high frequency swing due to simultaneous connection of all GFM WTs. Although this would be avoided in reality due to sequential connection of WTs, it gives a glimpse into the characteristics of the different control methods. The VSG and dPLL, with standard cascaded voltage-current control, show reduced damping at such low bandwidths, as discussed in section 4.2. Although the VSG has inertia that provides additional attenuation, it introduces a sub-synchronous resonance mode that can be seen in the later stages. The PSC has integral control in its voltage loop to suppress the high frequency disturbances along with a high pass filter for damping the resonant modes. Moreover, since the DPC has the standard vector current control structure, it has a well damped response.

In Stage 2, the event at 1.6 s is quite demanding in terms of the power peak for dPLL and DPC, and is associated with a large frequency swing, especially for dPLL due to reduced system damping from the low bandwidth cascaded controller. For the power-synchronization controls viz. PSC and VSG, their active damping characteristic or inertia helps slow down the swing. However, looking closely at 1.6 s (Stage 2) in Fig. 6, the effect of the right half plane zero, although well damped, can be seen in the non-minimum phase behaviour.

In Stages 3 and 4, the power-synchronization based methods viz. PSC and VSG have the largest swing in frequency and power due to the capacitors in the circuit being charged resulting in negative Var for the WPP and a changed coupling in the circuit. This shows that power-synchronization based methods are prone to oscillation when the system coupling is changed. Lastly the DPC has no closed loop voltage control but just standard current control with a virtual phase angle, and shows superior performance, compared to the other control schemes. This indicates that having decoupling feed-forward terms in the voltage loop can deteriorate the performance, especially during energization of capacitive loads that change the networks coupling in the initial stages of blackstart. However, although DPC works well in islanded mode, future studies are needed to investigate its behaviour during synchronization transients and faults.

Overall the VSG control shows oscillatory behaviour in all stages. This is due to the reduced damping in the system for low bandwidth cascaded voltage-current control. As discussed in section 4.2, although virtual resistance damps the synchronous oscillations, it changes the coupling leading to more resonance modes. Inertia has been tuned to provide additional damping, but it introduces a low frequency mode, which is excited in all the different stages in Fig. 6. Sun et al. (2019) shows that oscillation damping by modifying outer power control loops—using a derivative term for lead-lag controller, or with adaptive feed-forward compensation—can avoid requirement of virtual impedance and damp oscillations more flexibly. The frequency swing is most pronounced in Stage 4b at 2.8 s when the onshore MMC lower arm cells are pre-charged. This event also triggers a near 50 Hz oscillation for the dPLL due to the reduced damping, again as a consequence of the standard cascaded control structure with low bandwidths. These oscillations will be more damped for greater values of resistance in the system. It is important to note here that no auxiliary load has been simulated and the WT converter equivalent resistance is considered to be 0. Thus, system damping is limited only to losses and transformer resistance.

## 6 Conclusions

Recent field tests on HVDC interconnectors have shown that VSC-HVDC can be used for blackstart services. This makes VSC-HVDC connected offshore wind power plants promising candidates for providing blackstart and islanding operation capabilities, as conventional generation is being phased out and wind power plants grow bigger to meet the decarbonization aims. This paper presents an analysis of the transient behaviour of an HVDC-connected offshore wind power plant participating in a traditional bottom-up power system restoration procedure and focuses on grid-forming as the main control change required to enable blackstart and islanding services from wind turbines. The general working principle of grid-forming control has been explained with the constituent functional blocks, along with conceptual explanation of four different techniques viz. Virtual Synchronous Generator, Power Synchronization Control, Distributed-PLL based control and Direct Power Control. These methods were then implemented and compared in a study of the blackstart of onshore load by an HVDC-connected offshore wind power plant, focusing on transients due to energization of transformers, cables, MMC cells and HVDC link.

The simulation results show that all the four methods are able to deal with the energization transients in a controlled manner while maintaining stability of voltage and frequency at the offshore terminal. However, differences in their transient behaviours were observed and a qualitative discussion was presented. It has been shown that the low bandwidth of standard cascaded control structure—in VSG and dPLL—reduces system damping, pushing the system closer to instability. Moreover, the performance of power-synchronization based methods—viz. VSG and PSC—depends on the network coupling and can deteriorate for capacitive loads. Finally the lack of any decoupling terms in the voltage control—for PSC and DPC—results in a superior performance, as the network is weak, with less clear decoupling between the traditional $P - f$ and $Q - V$ inter-dependencies. While the results presented in this paper provide an initial comparison between the different grid-forming control strategies, further investigation is needed, especially in regards to harmonic load sharing, synchronization transients during sequential energization of wind turbines inside the wind power plant, and the effect of blackstart and islanded operation on rotor and turbine DC link dynamics, before concluding on the best control approach for black-starting the wind turbines. Lastly auxiliary load is expected to improve performance, by enhancing the damping in the system, which is critical in the initial stages of blackstart, and should be modelled.

## Appendix A: Parameters

*Code and data availability.* Inquiries about data and requests for access to the simulation models used in this study should be directed to the authors.

*Disclaimer.* This article is part of the special issue "Wind Energy Science Conference 2019". It is a result of the Wind Energy Science Conference 2019, Cork, Ireland, 17–20 June 2019.

**Table A1.** Main circuit parameters of the model [T2 - offshore, T1 - onshore, $X_L$ - leakage reactance].

| Parameters | Values |
|---|---|
| WT rating | 8 MW, 66 kV |
| WT GSC Filter | $L_f = 10\%, C_f = 5\%$ |
| WT transformer | 0.69/66 kV |
| | $R = 1\%, X_L = 1\%$ |
| WPP rating | 400 MW |
| HVDC transformers | 1200 MVA, $X_L = 15\%$ |
| | T2: 66/390 kV |
| | T1: 390/400 kV |
| PIR, PIT | 120 Ω, 0.3 s |
| HVDC link rating | ±320 kV, 1200 MW, 200 km |
| MMC | 1200 MVA |
| | 225 submodules per arm |
| Onshore load | 30 MW |

*Author contributions.*  JS designed the simulation for the case study. AJ and JS implemented the models and carried out the simulations. NC and AJ analysed the results. AJ prepared the manuscript with contributions from NC and JS.

*Competing interests.*  The authors declare that no competing interests are present.

*Acknowledgements.*  The authors gratefully acknowledge the contributions of Oscar Saborío-Romano to the discussions leading up to this work. This work is part of the InnoDC project that has received funding from the European Union's Horizon 2020 research and innovation programme under the Marie Skłodowska-Curie grant agreement No 765585.

**Table A2.** Tuned control parameter values in pu.

| Control | Parameters | Values |
|---------|-----------|--------|
| VSG | $P_i$ | 0.4 |
| | $PI_v$ | 0.11, 2.55 |
| | $r_v$ | 0.2 |
| | $J$ | $2 \times 10^3$ |
| | $D$ | $4 \times 10^5$ |
| | $k_q$ | 0 |
| PSC | $r_v, \alpha_v$ | 0.5, 40 |
| | $r_u$ | 0.8 |
| | $k_p$ | 0.004 |
| | $PI_q$ | 1, 0.01 |
| dPLL | $P_i$ | 0.4 |
| | $PI_v$ | 0.02, 2.55 |
| | $PI_{PLL}$ | 10, 25 |
| | $k_f$ | 50 |
| | $PI_p$ | 4, 40 |
| | $k_q$ | 0.01 |
| DPC | PI | 0.012, 0.21 |

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
