# Peer review of "Grid-forming control strategies for blackstart by offshore wind power plants"

_Wind Energy Science, 2020_

## Referee Comment (RC1) · Anonymous Referee #1 · 10 Apr 2020

This paper compares grid-forming strategies when used for blackstart by offshore wind turbines. The presentation of each control strategy is clear and the benchmark model to compare is suitable for the comparison. My main concern is about how dependant are the comparison results with the control parameters chosen by the authors. Also, the presentation of the general grid -forming scheme should be improved since this presentation is mixed with specific details of control strategies that are later explained again.

My specific comments are as follows: 1) In Section 3.1 the authors should clarify when PLL is used in the control scheme presented in Figure 1. At this moment, I understand that PLL is used during the grid-connected operation mode, but from the current explanation is not completely clear. 2) At the end of Section 3.1 (from sentence starting

in line 200) is it not clear if these alternative control structure would replace the secondary droops and power controllers shown in Figure 1 or just the secondary droops. 3) Some control schemes like power synchronization or emulation of swing equation are explained twice in Sections 3.1 and 3.2. Even in Section 3.2 they are presented as a general review and later with all details. To improve clarity of presentation I would present Figure 1 with general details about the function of each block. Then later, in Section 3.2 for each subsection I would introduce the fours control systems with both their general operation and the details of the control blocks. 4) How did the authors choose the control parameters to ensure that all control structures are compared under the same conditions? Could DPC and PSC tunning be changed to improve performance? 5) It is clear that frequency shows much clear differences among the control strategies. However, the general differences in terms of all magnitudes (P, Q, f and V) must be explained at the beginning and later explain more in detail the differences in terms of frequencies.

My comments about format are as follows: 6) The authors extensively use words in italic format. I would not use italic format in the paper unless there is an important reason. 7) Section 2.3 does not have title. 8) In Figure 2: - In some control schemes it is not completely clear how dq components are merged or split. - In VSG, it is not clear if the virtual impedance Zv is applied to d, q or both components. - d and q are very small in all subfigures. 9) Please indicate T1 and T2 in Figure 3.

---

## Referee Comment (RC2) · Anonymous Referee #2 · 4 May 2020

This paper includes a brief literature review of grid-forming converter control strategies and compares their performance when implemented on offshore wind farms. In general the paper is fairly written, however there is a very little novelty in the paper. My specific comments are as follows:

1. The Abstract and Introduction are well written and related literature has been identified and cited well. 2. Section 2.3 does not have a title and should be included. 3. In Line 142 the term Inertial is spelled wrong. Please do a proof reading of the paper. 4. In Fig. 1 which voltage is controlled by the Cv loop ? From my understanding it should be replaced by a block that generates current references from back calculation as in PSC. In PSC, The PCC voltage or reactive power is controlled using droop based or integral controllers (named as secondary droop in Fig. 1) which takes the input as

[Figure]

delta V or Delta Q. The structure shown in Fig. 1 is similar to VSG. However, in this case the input delta V to the secondary droop should not be present. Hence, I would suggest restructuring of this general block scheme and an improved explanation to it. 5. In Line 180 the word alternatively is spelled wrong. Please do a careful proof reading of the paper. 6. Figure 2c) is not correct. The output of AVC in PSC is NOT the reference voltage for the PCC, Vf*. Once the active damping has been implemented it gives the reference voltage for the converter in the converter DQ frame. This reference is used along with the PCC voltage and current measurements to estimate the real current which is governed by equation 7. If the estimate is greater than the threshold the current references are set using hard control and finally the current controller generates the new reference voltage for the converter in the converter DQ frame. Please correct it. I would also suggest that block scheme for all the control strategies should be modified. The vector and scalar quantities should be identified clearly. 7. Please show the tuning criteria for all the control parameters while explaining the control strategies. 8. In equation 3 dw/dt should not be equal to theta. Theta is integration of w over time. 9. Comparing DPC with other control strategies that include closed loop current controller seems unfair to me because the dynamics of closed loop current controller are not present in DPC. 10. Please mark terminals T1 and T2 in the figure 3.

The paper Italicize the text to a great extent. This is not required. Please italicize only important text.

---

## Author Comment (AC1) · 4 Jun 2020

Dear Reviewer,

The authors would like to appreciate your effort and strongly believe that your comments have truly helped to improve the quality and credibility of the manuscript along with its comprehensiveness.

The attached file contains the response to your specific comments with the revised manuscript as addendum for reference, colour coded for the different comments.

Thank you

Please also note the supplement to this comment:

[Figure]

https://www.wind-energ-sci-discuss.net/wes-2020-34/wes-2020-34-AC1-supplement.pdf

[Figure]

**Supplement:**

Dear Reviewer,

The authors would like to appreciate your effort and strongly believe that your comments have truly helped to improve the quality and credibility of the manuscript along with its comprehensiveness. The revised manuscript is attached herewith and has been revised with due consideration to all the reviewers' comments.

Given the extent of the changes implemented in the manuscript, we choose not to include the specific text in this document, but rather include it in the revised manuscript. A colour code has been used to address the specific comments: green for RC1, blue for RC2 and red for both.

The following major changes have been made in the revised manuscript.

The presentation of the paper has been modified:
1. Sections 1 & 2 have been re-organized, with a paragraph on the Contribution added to Section 1 (Introduction), and Section 2 re-titled as Literature Review.
2. Fig. 2 has been re-drawn for clearer comparison of the different control schemes.
3. A proof reading of the paper has also been done to remove unnecessary abbreviations and for consistency of word usage e.g. wind power plant instead of wind farm – also in the title.

The simulation results – and the consequent conclusions – have changed, due to two main reasons:
1. We have re-tuned the controllers to fit a more realistic scenario. The voltage controller, originally tuned for a bandwidth of 1.4 kHz, was very fast, being very close to the WT converters switching frequency of 1 kHz. In the revised manuscript, the voltage controller bandwidth has been set to 40 Hz. All the controller values are included in the appendix (Please refer to Table A2).
2. An error in the implementation of the DPC method was found and corrected, significantly improving its performance (Please refer to Fig. 6).

The response to your specific comments are given below.

| # | Comment | Response |
|---|---------|----------|
| | This paper compares grid-forming strategies when used for blackstart by offshore wind turbines. The presentation of each control strategy is clear and the benchmark model to compare is suitable for the comparison. My main concern is about how dependant are the comparison results with the control parameters chosen by the authors. Also, the presentation of the general grid -forming scheme should be | Thank you.
->Tuning criteria of the controllers has been added. Please refer to Section 4.2.

->The general grid-forming scheme (Section 3.1) has been simplified and reorganized along with the explanation of the different control strategies (Section 3.2), to take into account the comment. Please refer to Section 3. |

| | | |
|---|---|---|
| | improved since this presentation is mixed with specific details of control strategies that are later explained again. | |
| 1) | In Section 3.1 the authors should clarify when PLL is used in the control scheme presented in Figure 1. At this moment, I understand that PLL is used during the grid-connected operation mode, but from the current explanation is not completely clear. | Please refer to line 189 (Section 3.1) on Page 8. |
| 2) | At the end of Section 3.1 (from sentence starting in line 200) is it not clear if these alternative control structure would replace the secondary droops and power controllers shown in Figure 1 or just the secondary droops. | The explanation of the general structure has been simplified for more clarity. Specific details of the different control schemes have been removed.
Please refer to (end of) Section 3.1. |
| 3) | Some control schemes like power synchronization or emulation of swing equation are explained twice in Sections 3.1 and 3.2. Even in Section 3.2 they are presented as a general review and later with all details. To improve clarity of presentation I would present Figure 1 with general details about the function of each block. Then later, in Section 3.2 for each subsection I would introduce the fours control systems with both their general operation and the details of the control blocks. | The explanation of the general structure has been simplified for more clarity. Specific details of the different control schemes have been removed.
In Section 3.1 line 193 (Page 8), the swing equation is only meant to explain Power-based synchronization (from Synchronous generators) as an alternative to Voltage-based synchronization (PLL).
The general grid-forming scheme (Section 3.1) has been simplified and reorganized along with the explanation of the different control strategies (Section 3.2), to take into account the comment.
Lines 288-291 have been added to explain how each control scheme fits into the general control structure.
Please refer to Section 3 (text in green). |
| 4) | How did the authors choose the control parameters to ensure that all control structures are compared under the same conditions? Could DPC and PSC tunning be changed to improve performance? | Tuning criteria of the controllers has been added. Please refer to Section 4.2. Controller values are given in the appendix (Please refer to Table A2).
An error in the implementation of the DPC method was found and corrected, resulting in improved performance, as shown in Fig. 6. Please refer to Section 5 for discussion on the results. |

| 5) | It is clear that frequency shows much clear differences among the control strategies. However, the general differences in terms of all magnitudes (P, Q, f and V) must be explained at the beginning and later explain more in detail the differences in terms of frequencies. | Done.
Please refer to Section 5.
Lines 391-398 (Page 17): PQ explanation
Lines 399-412 (Page 17): V explanation
Lines 413-446 (Page 19): f explanation |
|---|---|---|
| 6) | The authors extensively use words in italic format. I would not use italic format in the paper unless there is an important reason. | Done. |
| 7) | Section 2.3 does not have title. | Done.
This section has been titled 'Contribution' and moved to Introduction.
Please refer to Section 1.2. |
| 8) | In Figure 2: - In some control schemes it is not completely clear how dq components are merged or split. - In VSG, it is not clear if the virtual impedance Zv is applied to d, q or both components. - d and q are very small in all subfigures. | Fig. 2 has been redrawn, clarifying vectors and scalars. Please refer to line 289 on Page 12. |
| 9) | Please indicate T1 and T2 in Figure 3. | Done.
Please refer to Fig. 3(a). |

Finally, we would like to thank you for your time and precise comments to the paper.
Kindly also note the supplement to this response – attached revised manuscript.

[revised manuscript text omitted]

---

## Author Comment (AC2) · 4 Jun 2020

Dear Reviewer,

The authors would like to appreciate your effort and strongly believe that your comments have truly helped to improve the quality and credibility of the manuscript along with its comprehensiveness.

The attached file contains the response to your specific comments with the revised manuscript as addendum for reference, colour coded for the different comments.

Thank you

Please also note the supplement to this comment:

https://www.wind-energ-sci-discuss.net/wes-2020-34/wes-2020-34-AC2-supplement.pdf

[Figure]

**Supplement:**

AUTHOR RESPONSE TO RC2

Dear Reviewer,

The authors would like to appreciate your effort and strongly believe that your comments have truly helped to improve the quality and credibility of the manuscript along with its comprehensiveness. The revised manuscript is attached herewith and has been revised with due consideration to all the reviewers' comments.

Given the extent of the changes implemented in the manuscript, we choose not to include the specific text in this document, but rather include it in the revised manuscript. A colour code has been used to address the specific comments: green for RC1, blue for RC2 and red for both.

The following major changes have been made in the revised manuscript.

The presentation of the paper has been modified:
1. Sections 1 & 2 have been re-organized, with a paragraph on the Contribution added to Section 1 (Introduction), and Section 2 re-titled as Literature Review.
2. Fig. 2 has been re-drawn for clearer comparison of the different control schemes.
3. A proof reading of the paper has also been done to remove unnecessary abbreviations and for consistency of word usage e.g. wind power plant instead of wind farm – also in the title.

The simulation results – and the consequent conclusions – have changed, due to two main reasons:
1. We have re-tuned the controllers to fit a more realistic scenario. The voltage controller, originally tuned for a bandwidth of 1.4 kHz, was very fast, being very close to the WT converters switching frequency of 1 kHz. In the revised manuscript, the voltage controller bandwidth has been set to 40 Hz. All the controller values are included in the appendix (Please refer to Table A2).
2. An error in the implementation of the DPC method was found and corrected, significantly improving its performance (Please refer to Fig. 6).

The response to your specific comments are given below.

| # | Comment | Response |
|---|---------|----------|
| | This paper includes a brief literature review of grid-forming converter control strategies and compares their performance when implemented on offshore wind farms. In general the paper is fairly written, however there is a very little novelty in the paper. | Thank you. A section has been added that highlights the Contribution of this paper. Please refer to Section 1.2. In general, this paper aims at covering the lack of literature comparing the different grid forming control strategies – typically developed for general purposes and driven by microgrid research – in a specific and demanding task as wind power plant providing black-start services. While non- |

| | | exhaustive, we think that this comparison will help direct future research in this area. |
|---|---|---|
| 1. | The Abstract and Introduction are well written and related literature has been identified and cited well. | Thank you. |
| 2. | Section 2.3 does not have a title and should be included. | Done.
This section has been titled 'Contribution' and moved to Introduction.
Please refer to Section 1.2. |
| 3. | In Line 142 the term Inertial is spelled wrong. Please do a proof reading of the paper. | Done.
Please refer to line 153 on Page 6.
A proof-reading of the paper has been done. |
| 4. | In Fig. 1 which voltage is controlled by the Cv loop? From my understanding it should be replaced by a block that generates current references from back calculation as in PSC. In PSC, The PCC voltage or reactive power is controlled using droop based or integral controllers (named as secondary droop in Fig. 1) which takes the input as delta V or Delta Q. The structure shown in Fig. 1 is similar to VSG. However, in this case the input delta V to the secondary droop should not be present. Hence, I would suggest restructuring of this general block scheme and an improved explanation to it. | Cv loop controls Vf, based on a reference. The current controller (sometimes only a limiter) aims at providing over current limitation. While standard cascaded V-I controller directly fits into the general structure shown in Fig. 1, the PSC structure does so also, with AVC (see Fig. 2(b)) & Iref generation (Eq. (5)) fitting into the blue box (voltage control, that generates the Iref for current control), followed by a standard current controller (the green box).
Please refer to Section 3.2.2 and Fig. 2(b).
The explanation of the general grid-forming structure and Fig. 1 has been simplified, along with corrections to Fig. 2. |
| 5. | In Line 180 the word alternatively is spelled wrong. Please do a careful proof reading of the paper. | Done.
Please refer to line 192 on Page 8.
A proof-reading of the paper has been done. |
| 6. | Figure 2c) is not correct. The output of AVC in PSC is NOT the reference voltage for the PCC, Vf*. Once the active damping has been implemented it gives the reference voltage for the converter in the converter DQ frame. This reference is used along with the PCC voltage and current measurements to estimate the real current which is governed by equation 7. If the estimate is greater than the threshold the current references are set using hard control and finally the current controller | Figure 2(b), now PSC (order of subfigs changed for consistency), has been corrected, with an improved explanation in Section 3.2.2.

The entire Fig. 2 has been re-drawn with better identification of vectors and scalars. Please refer to line 289 on Page 12. |

| | | |
|---|---|---|
| | generates the new reference voltage for the converter in the converter DQ frame. Please correct it. I would also suggest that block scheme for all the control strategies should be modified. The vector and scalar quantities should be identified clearly. | |
| 7. | Please show the tuning criteria for all the control parameters while explaining the control strategies. | Tuning criteria of the controllers has been added. Please refer to Section 4.2. Controller values are given in the appendix (Please refer to Table A2). |
| 8. | In equation 3 dw/dt should not be equal to theta. Theta is integration of w over time. | Done. Please refer to Eq. (1) on Page 10. |
| 9. | Comparing DPC with other control strategies that include closed loop current controller seems unfair to me because the dynamics of closed loop current controller are not present in DPC. | The DPC, as implemented in this study has the structure same as standard current control. This has been shown in the original reference, and re-iterated in explanation for DPC in Section 3.2.4 line 284 on Page 12. Moreover, an error in the implementation of the DPC method was found and corrected, resulting in improved performance, as shown in Fig. 6. Please refer to Section 5 for discussion on the results. |
| 10. | Please mark terminals T1 and T2 in the figure 3. The paper Italicize the text to a great extent. This is not required. Please italicize only important text. | Done. Please refer to Fig. 3(a). Done. |

Finally, we would like to thank you for your time and precise comments to the paper.
Kindly also note the supplement to this response – attached revised manuscript.

[revised manuscript text omitted]